# Socially Induced Infertility in Naked and Damaraland Mole-Rats: A Tale of Two Mechanisms of Social Suppression

**DOI:** 10.3390/ani12213039

**Published:** 2022-11-04

**Authors:** Nigel C. Bennett, Christopher G. Faulkes, Cornelia Voigt

**Affiliations:** 1Mammal Research Institute, Department of Zoology & Entomology, University of Pretoria, Pretoria 0084, South Africa; 2School of Chemical and Biological Sciences, Queen Mary College, University of London, Mile End Road, London E1 4NS, UK

**Keywords:** neuroendocrine, behavior, physiology, hormones, bathyergids

## Abstract

**Simple Summary:**

The naked and Damaraland mole-rats are group-living, subterranean mammals in which reproduction is distributed unequally among members of a social group, also referred to as reproductive skew. Only a single female per group, called the queen, produces offspring with the most dominant males of the group. The non-reproductive colony members are physiologically suppressed by the presence of the queen. This is reflected in their low concentration of luteinising hormone released from the pituitary and in their reduced responsiveness of the pituitary to stimulation with gonadotropin releasing hormone. Removal of the queen reverses these effects and leads to endocrine conditions in these females that are similar to those in reproductively active females. Regarding males, the extent of reproductive suppression is different between the two species. Non-reproductive male Damaraland mole-rats show hormonal profiles similar to the breeding males, whereas non-reproductive male naked mole-rats are physiologically suppressed similar to non-reproductive females. Thus, the two species represent ideal models to unravel the physiological, behavioural and neuroendocrine mechanisms regulating the hypothalamic-pituitary-gonadal axis. The recently discovered neuropeptides kisspeptin and RFamide-related peptide-3 are likely candidates to play an important role in the regulation of reproductive functions in the two mole-rat species.

**Abstract:**

The naked mole-rat (*Heterocephalus glaber*) and the Damaraland mole-rat (*Fukomys damarensis*) possess extreme reproductive skew with a single reproductive female responsible for reproduction. In this review, we synthesize advances made into African mole-rat reproductive patterns and physiology within the context of the social control of reproduction. Non-reproductive female colony members have low concentrations of luteinising hormone (LH) and a reduced response of the pituitary to a challenge with gonadotropin releasing hormone (GnRH). If the reproductive female is removed from the colony, an increase in the basal plasma LH and increased pituitary response to a GnRH challenge arises in the non-reproductive females, suggesting the reproductive female controls reproduction. Non-reproductive male Damaraland mole-rats have basal LH concentrations and elevated LH concentrations in response to a GnRH challenge comparable to the breeding male, but in non-breeding male naked mole-rats, the basal LH concentrations are low and there is a muted response to a GnRH challenge. This renders these two species ideal models to investigate physiological, behavioural and neuroendocrine mechanisms regulating the hypothalamic-pituitary-gonadal axis. The recently discovered neuropeptides kisspeptin and RFamide-related peptide-3 are likely candidates to play an important role in the regulation of reproductive functions in the two mole-rat species.

## 1. Introduction

The rodent moles of the family Bathyergidae provide an incredible opportunity to dissect the numerous reproductive strategies which are operative in this diverse family and that possess a broad spectrum of social organization [1]. Reproduction is a costly and lengthy process, requiring the growth and maturation of the gonads, courtship and copulation; followed by pregnancy and ultimately parturition in females, but it also is crucial for passing on copies of genetic material to the next generation. Several species of subterranean rodents are solitary, these animals are usually very territorial and exceptionally aggressive towards any conspecific or heterospecific mole-rats that enter the tunnel system [1]. The only time that pairing occurs is during the breeding season; outside of this period, any individuals entering the burrow system are usually killed [1]. Consequently, the majority of studies investigating the patterns of reproduction in solitary dwelling species of subterranean mammals have come from post-mortem studies in which specimens of both sexes have been collected on a monthly basis to investigate their reproductive condition. This usually involves the weighing and measuring of the dimensions of gonads, looking for the presence or absence of embryos, as well as assessing the numbers and types of follicles in the ovaries and the presence or absence of spermatozoa in the testes (see [2] review). In the current review, we synthesize the advances made into African mole-rat reproductive patterns and physiology with specific reference to two eusocial arid adapted species within the context of social control of reproduction.

## 2. Reproductive Scenarios in the Bathyergidae

Probably the most thoroughly investigated subterranean group of animals who have had their reproductive biology comprehensively evaluated is the African mole-rats, these. Studies have revealed a diverse array of reproductive strategies. This review highlights the advances that have been made in dissecting the patterns and trends exhibited in this remarkable family and, in particular, with reference to the mechanisms of social suppression that appear to be operating in two eusocial species for which we know the most.

The African mole-rats are a group of totally subterranean rodents that show a broad degree of social organization with diverse reproductive tactics [3,4]. The strictly solitary mole-rats are highly xenophobic and multiple occupancy of burrows only occurs for the briefest of periods during a defined breeding season [5,6,7,8]. In contrast, the truly social, or eusocial species have an extreme monopolization of reproduction by a single female in the colony who mates with between one to three male reproductive consorts that remain together for many years within the colony [9]. All offspring of the colony are philopatric to varying degrees, this being dependent upon the species, and can lead to the formation of large groups where dispersal is uncommon [10,11,12,13,14,15,16].

Solitary species endemic to South Africa and southern Namibia (*Georychus capensis* and *Bathyergus suillus* as well as *Bathyergus janetta*) occur usually in mesic habitats that have high rainfall that is predictable with a defined seasonal component with changes in photoperiod, ambient temperature and rainfall pattern [3,4,17]. The solitary eastern African mole-rat, *Heliophobius argenteocinereus,* inhabits relatively mesic regions, but this species breeds seasonally, cueing into the bimodal rainfall pattern specifically [7,8,18]. In the genus *Cryptomys*, all subspecies are colonial and live in regions of varying degrees of aridity ranging from the mesic western Cape through to the harsh, arid Kalahari in the northern Cape [16,19]. At the extreme end, or pinnacle, of social development are two species regarded as being truly social or eusocial mole-rats. These are found in semi-arid to arid habitats where rainfall tends to be highly unpredictable, or sporadic, with very brief windows of opportunity for mole-rats to extend their burrow systems in search of food (underground storage organs or geophytes) and to disperse when the soils are friable [2,17,20].

Solitary mole-rats are strongly territorial and aggressively defend their burrow system. The only time that multiple occupancy of a burrow system arises is when the mole-rats pair up, albeit briefly, to mate during the breeding season or when the female has young [5]. Mating is brought about by opposite-sexed individuals signaling to one another using their hind feet to drum, the frequencies of which vary with the sex of the animal. Generally, males signal with a faster drumming speed than females [5]. Once the female is receptive to the initiator, the male will tunnel into her burrow, court her briefly and mating will result. In both species of *Bathyergus* and the monotypic *Georychus*, the males burrow towards the burrow system of the female, whereby if the female is receptive, the male will mate multiple times and then leave the female burrow system by sealing off a section of the female’s tunnel and digging towards its own burrow system. In *Bathyergus janetta*, it has been reported that males may venture onto the surface and dig down into fresh mounds of the female’s burrow system to enter and mate with the female. Footprints can be found around fresh mounds (M. Herbst, Pers. Obs.), suggesting the males come onto the surface. Multiple, but brief, copulations result, following which the female remains on her own in the burrow system to give birth and rear the pups.

Subterranean rodents occur in a constrained environment where the burrow microhabitat in which the mole-rats inhabit is hypoxic, hypercapnic, devoid of light and experiences a muted temperature regime when compared to the surface [2,21]. In seasonally breeding rodents that occur above ground, reproduction has been primarily attributed to changes in proximate factors such as photoperiod and temperature changes. However, in mammals that spend the majority of their life underground, where there are no light cues and where animals possess microphthalmic eyes, other proximate environmental cues may be important, such as burrow temperature changes and the moisture content of the soil. Rainfall might have a more crucial role in triggering reproduction with the production of offspring confined to a specific period in strongly seasonally breeding species [4]. Herbst, et al. [6], found a strong relationship between rainfall and an increase in reproductive sex hormones of both sexes in the Namaqua dune mole-rats, *Bathergus janetta,* suggesting rainfall may be the environmental cue for the onset of mating. This observation is also supported from a calendar year study on a population of Cape dune mole-rats, *Bathyergus suillus,* from the western Cape, where the onset of the rains was found to coincide with the initiation of the breeding season [22]. More recently, Katandukila, et al. [23], and Ngalameno, et al. [8], have also shown in the African spalacid, *Tachyoryctes spendens,* and Emin’s mole-rat, *Heliophobius emini,* from East Africa, reproduction is cued to the bimodal rainfall pattern experienced in Tanzania.

In the social mole-rats, the environmental factors that are used to cue reproduction appear to be far less clear. Mole-rats in the genera *Heterocephalus* and *Fukomys* reproduce aseasonally and produce offspring throughout the year [4,24,25,26]. In the genus *Cryptomys*, two subspecies, the common mole-rat, *Cryptomys h. hottentotus,* and the highveld mole-rat, *Cryptomys hottentotus pretoriae*, [16,19,27] exhibit strongly seasonal reproduction, while two other subspecies, the Natal mole-rat, *Cryptomys h. natalensis,* and the Mahali mole-rat, *Cryptomys h. mahali,* are aseasonal breeders [28,29,30]. In the genera *Heterocephalus* and *Fukomys,* reproduction appears to be independent of rainfall, whereas in the genus *Cryptomys*, rainfall appears to be crucial in shaping the seasonality of reproduction in the common and Highveld mole-rats.

The three social genera *Cryptomys, Fukomys* and *Heterocephalus* occur in extended familial groups. The family group comprises dominant reproductive parents and several philopatric litters of non-reproductive offspring [11,13,15,31,32,33,34].

Within the genus *Fukomys,* and particularly the Damaraland mole-rat, *Fukomys damarensis*, colonies may be 40 individuals strong, but this is rare, and characteristically, colonies are more commonly composed of around 10–12 individuals. In the naked mole-rat colony, size is usually around 60–70 individuals, but in an agricultural field a colony of 290 individuals was captured [4,13,35].

The use of microsatellite markers in the Damaraland mole-rat has shown that males may enter colonies, mate with the female breeder, and then subsequently exit the colony. It is possible that the same modus operandi arises in the naked mole-rat, as dispersers of both sexes have been recorded from extensive field studies [36,37]. Regrettably, there is no published literature on the parentage in wild naked mole-rat colonies. In the naked mole-rat, a distinct phenotype arises in colonies. This phenotype is morphologically, behaviorally, and hormonally distinct [38]. The dispersers possess elevated concentrations of circulating luteinising hormone (LH), which are greater than in non-breeding males. The disperser phenotype is reproductively primed to breed following dispersal, yet they do not engage in reproduction in their natal colony. Colonies of the Damaraland mole-rat also possess a physiological group of larger, non-breeding animals that, instead of contributing to colony work, rather build up fat reserves in order to disperse and set up new colonies by pairing with unrelated individuals of the opposite sex when environmental conditions become optimal, such as a number of good rainfall events [39]. The naked and Damaraland mole-rat both possess dispersal phenotypes characterized by having fat reserve and performing less burrow maintenance work while resident within colonies. Following a significant rainfall event, the large, lazy workers either undergo dispersal and establish new colonies, or move into established colonies [38,40]. All social mole-rats share a common pattern of colonial living with reproduction monopolized by a single reproductive queen, but the way in which reproductive inhibition is enforced on the non-breeding colony members is divergent between species [9,39].

## 3. The Scenario in the Naked Mole-Rat

In the naked mole-rat, reproductive suppression occurs in both non-breeding males and females, the members of which remain philopatric to the colony and are physiologically suppressed from reproducing [9,41,42,43]. Socially induced infertility in non-breeding males is unique and has currently not been reported in any other cooperatively breeding mammal [9,43]. 

The ovaries and uterine horns of non-breeding female naked mole-rats are much smaller than their breeding counterparts, and their ovaries show little follicular activity when compared to the ovaries of the breeding females. Their ovaries are pre-pubescent in appearance, with few, if any, follicles [44]. While the majority of social mole-rat species are induced ovulators, the naked mole-rat is a spontaneous ovulator [42]. Endocrine studies support the findings on ovarian activity, such that the non-breeding females have low, or non-detectable urinary oestrogen and progesterone concentrations, and furthermore, no cyclical peaks in progesterone occur, indicative of anovulation [41,45]. The absence of a follicular cycle and subsequent ovulation appears to be due to very low concentrations of circulating LH arising from the anterior pituitary. Reproductive suppression in non-breeding naked mole-rats is believed to be the consequence of a reduction or disruption of GnRH release or potentially down regulation of GnRH receptors on the pituitary itself [43]. The ovaries of non-breeding female naked mole-rats are prepubescent while these animals remain in the colony with the breeding female. If a female is taken out of the colony and housed or paired with a male ovarian activation arises with the subsequent reinstatement of follicular development and ovulation [41]. Ovarian activity in a non-breeding female can result from a challenge to the breeding female from a high-ranking subordinate female in a colony or following dispersal and outbreeding in the field [38]. In captivity, a new breeding female frequently arises from in the natal colony through succession as a result of social queuing. Non-breeders may be more reproductively primed than other subordinates and challenge the breeding female, or even kill the existing breeding female, even if this involves mating with an adult sibling [46]. Laboratory studies have reported that an outbreeding reproductive strategy is the preferred option in this species [47,48], and this has been confirmed from studies of wild colonies in the field [38].

The neuroendocrine mechanisms leading to the onset of reproductive activity in female subordinate naked mole-rats are still not well-understood. In spontaneously ovulating mammals, the positive feedback effect of increased ovarian oestradiol production leads to GnRH secretion from the hypothalamus, which stimulates the preovulatory LH surge from the anterior pituitary. The neuropeptides kisspeptin (encoded by the Kiss1 gene) and RFamide-related peptide-3 (RFRP-3, encoded by the Rfrp gene) constitute important regulators of GnRH release, but with opposing effects. Kisspeptin activates GnRH neurons and is essential for normal reproductive activity and the timing of puberty onset. RFRP-3, on the other hand, is thought to be the mammalian homolog of the gonadotropin-inhibiting hormone (GnIH) in birds, which inhibits GnRH neuronal activity and gonadotropin release for review, see [49]. Kisspeptin neurons reside in two main regions of the hypothalamus: the anteroventral periventricular nucleus (AVPV) and the arcuate nucleus (ARC). In female rodents, oestradiol stimulates kisspeptin neurons in the AVPV while those in the ARC are inhibited. This positive and negative feedback effect of oestradiol is mediated by oestrogen receptor α (ERα), which is co-expressed in both kisspeptin neuron populations [50]. Kisspeptin neurons in the AVPV send projections to GnRH neurons in the rostral preoptic area and constitute the target for oestradiol to induce the preovulatory GnRH/LH surge [51,52]. The characterisation of these neuropeptides opens up a whole new field of research for investigating the control of reproduction in socially suppressed mammals. However, in naked mole-rats, only a few investigations have been undertaken to date. Zhou, et al. [53], have found that non-breeding female naked mole-rats possess significantly fewer kisspeptin immunoreactive cells in the AVPV when compared to breeding females. No such differences occur in the ARC. This suggests as with other female rodents, the AVPV neuron population is involved in the pre-ovulatory LH surge in naked mole-rats. Moreover, it implies that triggering the activation of these kisspeptin neurons is essential for the onset of reproductive activity. However, more recent studies have found evidence that RFRP-3 plays a crucial role in suppressing puberty onset [54]. 

Reproductively quiescent females show increased RFRP-3 expression in the dorsomedial hypothalamus when compared to reproductively active females. Furthermore, exogenous RFRP-3 prevents puberty onset in subordinate females that are removed from their natal colony and therefore given the opportunity to enter puberty [55]. Using quantitative PCR, Faykoo-Martinez, et al. [54], have identified several candidate genes, including those of the kisspeptin signaling pathway, which show a differential expression pattern in relation to reproductive status. However, more research is needed to identify the exact roles of RFRP-3 and kisspeptin on the activation of the HPG axis in reproductively quiescent females.

The breeding and non-breeding males from both captive and wild colonies of the naked mole-rat show significant differences in both the size and appearance of the reproductive tracts [56]. Breeding males have larger testes relative to their body mass compared to their non-breeding counterparts. While all males show spermatogenesis and mature sperm production, the non-breeders produce significantly fewer sperm compared to breeders. Furthermore, the spermatozoa in non-breeders appears to have larger numbers of sperm that are non-motile or possess morphological defects such as possessing two flagella, being double headed or pin headed [56]. Non-breeders characteristically have low concentrations of urinary testosterone and lower or non-detectable concentrations of basal circulating luteinising hormone (LH). Furthermore, the pituitary gland is less responsive to the administration of exogenous GnRH when compared to plasma levels of the breeding males [42]. These profiles imply that socially induced impairments occur on the hypothalamic-pituitary axis in non-breeding males [42,57]. However, in contrast to females, no reproductive status-related differences in the number of kisspeptin-expressing cells exist in the hypothalamus [53]. Nevertheless, subordinate males have increased RFRP-3 immunoreactivity in the hypothalamus and when given the opportunity, they do not enter puberty when treated with exogenous RFRP-3, which matches the findings in females [55]. This suggests that a similar mechanism of puberty suppression is acting in both sexes. Under natural conditions, reproductive suppression in males is lifted once the social environment is changed. As a consequence, non-breeding males removed from the inhibitory cues of the natal colony and either housed singly or paired with a female rapidly demonstrate elevated levels of plasma LH and urinary testosterone concentrations [57]. 

## 4. The Scenario in the Damaraland Mole-Rat

Incest avoidance appears to be the predominant mechanism of reproductive inhibition in the majority of the social *Cryptomys* and some members of the genus *Fukomys*, and is important in maintaining reproductive skew in *F. anselli* [58], *F. darlingi* [59,60], *F. damarensis* [61,62] and *F. mechowii* [63]. 

The ovaries of non-breeding female Damaraland mole-rats are more functionally active compared to those of non-breeding naked mole-rats, in that they exhibit a range of follicular development from primordial follicles through to Graafian follicles, with the latter regressing to form luteinised, unruptured follicles [64,65]. Being induced ovulators, corpora lutea of ovulation are absent in non-breeding females and concentrations of urinary progesterone do not reach the concentrations of those in breeding females [66,67]. The disruption of GnRH release or a down regulation of GnRH receptors on the pituitary itself are possible scenarios for the reduced response of the pituitary to an exogenous GnRH challenge [68]. The measurable progesterone found in the non-breeding females is most likely a function of the luteinisation of unruptured tertiary and Graafian follicles [64,68] At the neuroanatomical level no differences are found in the numbers of GnRH immunoreactive neurons, the different proportions of neurons (non-polar, unipolar and bipolar) or the size of the cell soma of the GnRH neurosecretory cells between breeding and non-breeding females [69]. However, the GnRH neurons of non-breeders retain more GnRH in the dendrites as a consequence of the lack of a signal to release into the portal system. The GnRH concentrations in the median eminence and proximal pituitary stalk are significantly higher in non-breeding females when compared to those of breeding females [69]. These findings support the lack of a preovulatory GnRH surge and subsequent ovulation in these females. 

In induced ovulators, the mating stimulus activates kisspeptin neurons in the AVPV, which leads to activation of GnRH neurons and consequently, to ovulation. No such activation occurs in the ARC kisspeptin neurons, indicating that this neuron population is not involved in mating-induced ovulation [70]. The release of GnRH is under the control of positive and negative feedback mechanisms of 17β-oestradiol, mediated by the oestrogen receptor α (ERα). Kisspeptin neurons in the AVPV and arcuate nucleus are known to co-express ERα and androgen receptors (AR). Several recent studies in female Damaraland mole-rats have identified differential hypothalamic gene expression patterns of steroid hormone receptors and of neuropeptides involved in the activation of GnRH neurons according to the reproductive status of the female. In breeding females, the mRNA expression of AR is significantly elevated compared to non-breeding females in several hypothalamic and limbic brain regions such as the medial preoptic area, the principal nucleus of the bed nucleus of the stria terminalis, the ventromedial nucleus of the hypothalamus, the ARC and the medial amygdala [71]. These findings could relate not only to the reproductive activity of these females, but also to their dominant position within the colony. Furthermore, breeding females have increased ERα expression in the AVPV, which is in agreement with the stimulatory effect of oestradiol on the kisspeptin neuron population in this region [71]. Interestingly, in Damaraland mole-rats, *Kiss1*-expressing cells within the preoptic hypothalamus are scarce, with only few cells scattered throughout the AVPV and the periventricular preoptic nucleus [72]. It is likely that in breeding females, these cells only become activated after the mating stimulus has occurred and ovarian oestradiol production has started to increase as has been shown in another induced ovulator, the musk shrew (*Suncus murinus*, [70,73]). In contrast, in the ARC, Kiss1 gene expression differs according to females’ reproductive status. These kisspeptin neurons coexpress neurokinin B (NKB; encoded by the Tac3 gene) and the endogenous opioid peptide dynorphin (encoded by the Pdyn gene) and are termed the “KNDy” (kisspeptin/neurokinin B/dynorphin) neurons. This neuron population is considered to be the GnRH pulse generator, which is responsible for generating the pulsatile release of GnRH. According to the model, increased NKB expression signals pulse onset. The release of NKB leads through a positive feedback loop to increased neural activity of the KNDy neurons, followed by kisspeptin release and GnRH secretion. The subsequent release of dynorphin terminates the kisspeptin release and the GnRH pulse (for review, see Moore [74]). Female breeding Damaraland mole-rats have significantly more *Kiss1*-expressing cells, increased neurokinin B and decreased dynorphin gene expression in the ARC compared to non-breeding females [72,75]. These differential gene expression patterns according to reproductive status are in line with the finding that in all mammals, the GnRH pulse generator reactivates at puberty after juvenile quiescence (for review, see [76]). In induced ovulators, such as the musk shrew, virgin mating activates ovarian steroidogenesis, thereby priming the HPG axis and consequently, inducing the onset of puberty [77]. Therefore, the hypothalamic gene expression pattern found in non-breeding female Damaraland mole-rats most likely reflects their pre-pubertal stage. Similar to the findings of [55] in naked mole-rats, female breeding Damaraland mole-rats had significantly fewer RFRP-3-expressing cells within the hypothalamus than non-breeding females [72]. Although the exact mechanisms of puberty onset are still unknown and may differ between the two species, this finding supports the view that RFRP-3 plays an important role in it.

In male Damaraland mole-rats, there are few anatomical differences between the reproductive tracts of breeders and non-breeders, with no differences in the production of spermatozoa or any defects in motility or morphology in non-breeding males [56,78]. However, breeding male Damaraland mole-rats, as with the naked mole-rat, have heavier testes relative to their body mass. Interestingly, larger testes do not result in an enhanced production of spermatozoa, since no significant differences are found in sperm numbers between the two groups [56].

Endocrine studies complement these anatomical findings, in that both the breeding and non-breeding males do not differ in their concentrations of urinary testosterone, brain GnRH concentrations or basal circulating LH concentrations, and they have a similar pituitary response to an exogenous GnRH challenge [66,67,68,69]. Furthermore, there is no difference in the circulating FSH concentrations between the reproductive categories. However, breeding males have been shown to possess six times more FSH receptors in the testicular tissues when compared to those of non-breeding males [79]. Thus, the effect of FSH suppression in this instance must be at the level of the receptor, or the post receptor. 

Similar to females, neural gene expression patterns of steroid hormone receptors and neuropeptides involved in activation of GnRH neurons show differences in relation to reproductive status. Breeders have significantly increased androgen receptor and progesterone receptor expression in several hypothalamic brain regions involved in reproduction [80]. Furthermore, while the number of *Kiss1*-expressing cells in the ARC is similar in breeders and non-breeders, the latter exhibit increased RFRP-3 gene expression in the dorsomedial hypothalamus [81]. This finding matches the recent findings by immunocytochemistry in male and female naked mole-rats [55]. It suggests that not only female, but also male Damaraland mole-rats are subject to physiological suppression.

## 5. The Role of Prolactin in the Suppression of Reproduction

Prolactin is important in mammalian sociality and in particular cooperative breeding. Prolactin has been termed the hormone of incubation in birds [82]. It can act as a double-edged sword, in that elevated prolactin is involved in the expression of social and cooperative behaviours in vertebrates, particularly so in birds and mammals [83,84], but it can also bring about suppression of reproduction in higher organisms [85]. Elevated concentrations of prolactin may bring about infertility in mammals since prolactin inhibits the secretion of luteinising hormone and follicle stimulating hormone and consequently brings about a form of natural suppression of reproduction usually arising during lactation [86].

The patterns of plasma prolactin have been found to be very different between the naked mole-rats and Damaraland mole-rats. As one would expect for a reproductively active female mammal that undergoes pregnancy and parturition, prolactin was detected in the breeding females of both the naked and Damaraland mole-rat with high values during pregnancy and subsequent lactation after birth of the offspring. Most non-breeding naked mole-rats of both sexes had detectable levels of prolactin, and some animals were even considered clinically hyperprolactinaemic [39]. In marked contrast, most non-breeding Damaraland mole-rats had undetectable levels of prolactin. The findings imply that while elevated prolactin in non-breeding naked mole-rats may be important in the evolution of social suppression, this is definitely not the case for the Damaraland mole-rat.

Both non-breeding male and female naked mole-rats have enlarged teats despite the fact that the latter do not become fertilized or pregnant (N.C. Bennett and J.U.M. Jarvis pers. obs). These elevated levels of prolactin observed in non-breeding naked mole-rats may bring about the enlargement of the teats, but the prolactin may also function to inhibit the release of GnRH and subsequently LH, FSH and their resulting target steroids. This in turn would lead to the block of follicular development in the females and spermatogenesis in males. It is also very tempting to speculate that the elevated prolactin in non-breeders of both sexes may encourage cooperative behaviours to develop towards different colony members. In marked contrast, the non-breeding animals of both sexes in Damaraland mole-rats lacked elevated circulating prolactin levels, yet they still express cooperative behaviour to some extent [87]. Nevertheless, there are elevated levels of prolactin receptor gene expression in the choroid plexus and in the arcuate nucleus of non-breeding female Damaraland mole-rats when compared to that of breeding females, which suggests higher local prolactin levels in the brain. Such elevated brain prolactin levels could inhibit kisspeptin neuron activity in the arcuate nucleus and thereby contribute to the reduced activation of GnRH neurons and reduced LH release from the pituitary [67,88]. Kisspeptin expression in the arcuate nucleus is known to be suppressed by elevated prolactin levels such as during lactation [89].

## 6. Future Directives in Assessing How Socially Induced Infertility Is Directed towards the Non-Breeders of Colonies

The most pressing question to answer is how the reproductive female manages to orchestrate reproductive suppression in the non-breeding female and male members of the colony. Further examination into the ways that neurobiological transmitters are involved in the stimulation and release of GnRH and those that are involved in its inhibition could allow us to assess how the breeding female is able to manipulate the hypothalamo-pituitary gonadal axis of her loyal colony members. A comparative neurobiological study is currently underway to investigate how social suppression is brought about in philopatric non-breeding animals within the group. Indeed, following the various pathways that may be involved in social suppression in the social brain may enable us to discover how reproductive control is maintained by the breeding female or “queen” in these mammals that show so many parallels to social insects. In conclusion, we have managed to unravel the mechanistic pathway of how social suppression arises in the non-breeding members of the colony, but we do not currently know how the female achieves this. In the naked mole-rat, it would appear that dominant control by the reproductive female arises by aggressive interactions directed towards the non-breeding animals that pose the most direct threat (this is achieved by shoving and pushing by the queen [90]). However, in the Damaraland mole-rat is would appear that self-restraint by the non-breeders is operational by a strong incest avoidance component, yet there also appears to be a physiological component about which we know little currently [61].

## Data Availability

Not applicable.

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
