# Peer review of "Socially Induced Infertility in Naked and Damaraland Mole-Rats: A Tale of Two Mechanisms of Social Suppression"

_animals, 2022, doi:10.3390/ani12213039_

Round 1
Reviewer 1 Report
I have now completed reviewing the article “Socially induced infertility in Naked and Damaraland mole-rats: a tale of two mechanisms of social suppression”. The current review paper highlights the advances made into African mole-rat reproductive patterns and physiology. There were no major concerns in the manuscript. I just pointed out several comments and suggestions which may improve the quality of the manuscript.
Please make sure that the introduction section is completed.
Please carefully check the layout and font of the manuscript. For example, lines 344-355, page 8.
There are fewer latest Refs cited in the current manuscript.
Author Response
Dear Reviewers, Thank you kindly for the suggested changes. I have made all the necessary changes to the manuscript as requested in the pdf citations have been added, scientific names provided as well as a conclusion at the end of the review. In the simple summary and abstract an explanation as why the review was undertaken has been provided. The language has been made more objective and subjective language removed. The paper has been formatted for 'Animals'. Thank you most kindly for the time and effort afforded to the paper. I sincerely hope it is now suitable for publication.
All overlaps have been changed into new text . The manuscript is now formatted for animals. Many of the citations are self citations this is because the majority of the research in this field has been done by the three authors, Bennett, Faulkes and Voigt. The review was undertaken to tie together older and more recent research into this field in the social arid adapted mole-rats.
Reviewer 2 Report
Dear Authors,
Thank you for submitting this paper that explores the Damaraland and naked more rat reproductive physiology. While a very interesting review, there appear to be multiple self-citations throughout the work, often to an extreme level. There is also a limited attempt to format the work to the Animals author guidelines. While the review in and of itself is interesting, there are numerous references to existing reviews, also written by the authors, so there needs to be a much clearer explanation as to what is novel about this study.

Author Response

(The authors gave the same response as above.)

Round 2
Reviewer 2 Report
Dear Authors,
Thank you for your clear revisions to the manuscript, and the explanations regarding citations. The revisions show a clear address of all key points.